# Identification of Reference Buildings in Mediterranean Countries: The HAPPEN Project Approach

**Manuel Rosales [1], Chrysanthi Efthymiou [2,*], Nikolaos Barmparesos [2], Panagiotis Tasios [2], José Manuel Salmerón Lissén [1] and Margarita Niki Assimakopoulos [2]**

[1] Grupo de Termotecnia, Escuela Técnica Superior de Ingeniería, Universidad de Sevilla, Camino de Los Descubrimientos S/n, E41092 Sevilla, Spain; rosalesgarciamanu@gmail.com (M.R.); jms@us.es (J.M.S.L.)

[2] Department of Applied Physics, Faculty of Physics, University of Athens, Building Physics 5, University Campus, 157 84 Athens, Greece; nikobar@phys.uoa.gr (N.B.); panagiotistasios@gmail.com (P.T.); masim@phys.uoa.gr (M.N.A.)

\* Correspondence: c-efthymiou@phys.uoa.gr; Tel.: +30-210-727-6731

**Abstract:** This study's scope is to collect and analyze all the needed information related to the residential building stocks in Mediterranean countries, especially those that participated in the framework of the HAPPEN project (Greece, Croatia, Cyprus, Italy, Slovenia, Spain, and France). A specific procedure was followed in order to conduct a coordinated evaluation of the residential building stock. The most important variables for a statistical examination of the building stock are outlined, as well as an approach for establishing reference buildings. National data for the seven participating nations were collected and evaluated using the prescribed methodology. The research findings identify six distinct reference buildings in each nation. More specifically, the most representative buildings were distinguished through a cross-country comparison of the obtained data, after classifying the buildings into different classes to which the same approach for deep renovation/refurbishment can be applied.

**Keywords:** reference buildings; building typology; deep renovation; Mediterranean zone; HAPPEN project

## 1. Introduction

The European Union aims to be climate-neutral with net-zero greenhouse gas emissions (80–95%) by 2050. Buildings with higher energy efficiency and the utilization of renewable energy sources in existing and new structures are projected to play a significant role in accomplishing this goal. Globally, buildings account for around 35% of resources, 40% of energy use (the 27% addressed to households), consume 12% of the world's drinkable water, and produce almost 40% of global carbon emissions.

The Energy Performance of Buildings Directive—EPBD [1] requires Member States (MSs) to specify minimum energy performance standards for buildings and building components in order to achieve cost-optimal levels and have a real impact on the reduction of building energy consumption. Ballarini et al. (2014) [2] define "cost-optimal" energy performance as energy performance with the lowest cost throughout the predicted economic lifespan. Simultaneously, it broadens the definition from cost-optimal to cost-effective.

Due to the wide range of different building typologies and HVAC systems, it is not possible to compute the cost-optimality for each individual building. As a result, the EPBD's comparative framework demands the MSs identify a set of reference buildings (RBs) that reflect the national or regional structures.

According to Annex III of the EPBD recast [3], RBs are "buildings characterized by and representative of their functionality and geographic location, including indoor and outdoor climate conditions", and are intended to represent the typical and average building stock

in terms of climatic conditions and functionality (e.g., residential buildings, schools, etc.). According to the EPBD's principles, "the basic objective of an RB is to depict the standard and average building stock in a certain MS". As a result, the RBs should be developed to reflect the actual national building stock as closely as possible in order to guarantee that the findings are representative.

The term "reference building" (RB) is not well defined or harmonized among the Member States. The Member States must define "reference buildings" that should represent the typical and average building stock in each Member State in order to obtain general results consistent with the characteristics of the analyzed building stock, according to the Commission Delegated Regulation No.244/2012 [4] and its accompanying Guidelines [5]. This information is also highlighted in the research of Touloupaki and Theodosiou (2017) [6].

As a result, a number of studies are being conducted at the EU and worldwide levels to define RB. Despite the fact that there is no standard for determining RBs, most researchers use comparable methods.

For example, Balaras et al. (2007) [7] and Dascalaki et al. (2010) [8] examined Hellenic residential buildings. The categorization was performed by using the year of construction (before 1980, 1981–2001, and 2002–2012), building typology (low-rise structures with one or two stories, high-rise structures with more than two stories), and climatic zone (four zones). Additional sub-categories were created based on shared features such as the building's thermal characteristics and HVAC systems. The process used to create these residential building categories assigns each one an actual existing building that is considered typical of all structures in the same category. Theodoridou et al. (2011) [9], on the other hand, focused on the stock of Hellenic households. The five proposed classes are class A (1919–1945), class B (1946–1980), class C (1981–1990), class D (1991–2010), and class E (2010–2011). This decision is driven by the fact that the building's age offers further information about the building's typologies, materials, supply systems, and appliances employed, as well as the construction method used. Similar characterization approaches were used by other researchers in countries such as Germany and Switzerland [10].

Balaras et al. (2005) [11] conducted a more comprehensive study of the energy usage of 193 residential building stocks from five European nations. The Danish residential building stock was studied by Tommerup and Svendsen in 2006 [12]. They were referring to two different types of structures: a single-family house and a multi-family one.

Uihlein and Eder (2009) [13] investigated European (EU27) residential building stocks, offering a model to depict the evolution of the respective building stocks for each of the nations. Three types of buildings have been recognized in further detail: single-family, multi-family, and high-rise. From 1900 until 2060, these classifications were further separated into historical and modern buildings.

The energy performance of residential building stocks in two big Italian regions, Piedmont and Lombardy, was studied by Fracastoro and Serraino in 2011 [14]. The study was conducted using data from the Italian census, which included 72 distinct building geometries, four different construction age categories, eleven different heating system efficiencies, and a variable number of degree-days (DD) categories with a predetermined step of 100 DD. Brandao de Vasconcelos et al. (2015) [15] established an approach for defining RBs in Portugal that suit the requirements and address the information gap. This strategy was successfully adopted in order to identify an RB (family home) that was typical of residential structures built in Lisbon between 1961 and 1990. In this study, building function type, building location, and age of construction were prioritized over other criteria used by existing works.

In Florianópolis, Brazil, Aline Schaefer and Enedir Ghisi (2016) [16] devised a strategy for generating RBs for low-income residential properties. Field data were collected in terms of creating a database of housing geometrical characteristics. Cluster analysis yielded two RBs: a 76-square-meter home with a living room, kitchen, and three bedrooms; and a 37-square-meter house with an integrated living room and kitchen, and two bedrooms.

Simulations have demonstrated that the RBs can accurately reflect their cluster, with degree-hour values that are comparable to the housing median sample.

A comprehensive survey was conducted by the Buildings Performance Institute Europe (BPIE) throughout all EU Member States [17]. This study gives an overview of the building stock in Europe. Building typology (function type), building age, building size, and building location were all factors in BPIE's survey. This categorization relates to the statistical data supplied by the EU nations that took part in the survey.

Furthermore, many European projects, such as TABULA [18], ASIEPI [19], and IMPRO–Building [20], have focused on the identification of RBs, either for promoting energy-efficient guidelines or validating tools and measures for energy performance requirements or facilitating cross-country comparison of building stocks.

According to Deru et al. (2011) [21], the Department of Energy (DOE) produced 16 RB models that characterize more than 60% of the commercial building portfolio in the United States. These models are intended to portray building attributes and construction techniques in a realistic manner. They were divided into three construction eras and consisted of 15 business buildings and one multi-family residential structure (pre-1980, post-1980, and new buildings). The information gathered for the creation of RBs can be divided into four categories that comprise a broader range of features: building type and general geometry, construction technologies and materials, HVAC systems and renewable energy production, and operational parameters. The study was conducted to evaluate new technologies, optimize designs, analyze advanced controls, produce energy codes and standards, and perform lighting, daylighting, ventilation, and indoor air quality research.

More recently, the research of Li et al. (2018) [22] proposed a building clustering methodology based on satellite images from Yuzhong district, China. The researchers claim that the K-medoids technique performed the most accurate results. The publication of Bhatnagar et al. (2019) [23] introduced a grouping model for reference Indian offices with the use of data collected from 230 buildings constructed in last decade. Four scenarios were developed according to energy consumption levels of those buildings. Moreover, the paper of Ledesma et al. (2021) [24] refers to a bottom-up model based on real-reference buildings as the first urban energy demand study concerning educational building stock in Quito, Ecuador, and Barcelona, Spain. Finally, Foroushani et al. (2022) [25] highlighted significant discrepancies in the North American Reference Building Approach and proposed the discontinuation of this method.

Based on the foregoing, we infer that the technical/scientific community has used several criteria to define samples for RB characterization. However, the criteria for building-stock energy performance may be summarized as being based on three factors: the climatic zone (region), geometry (footprint shape, dimensions), and thermophysical characteristics of the structures (U-values of enclosure). Thus, the main target of this paper is to classify the residential buildings of the Mediterranean area, based on the aforementioned criteria.

## 2. Methodology

In terms of climate zones, building styles, and usage, Europe's building stock is quite diverse. In fact, even within the same category, building usage might vary significantly between Member States. Climate conditions have an impact on construction methods and the energy requirements of a building. More specifically, the information gathered for the creation of RBs can be conveniently divided into four categories: 1. form: building type (e.g., office, school), size, and overall geometry of the structure; 2. envelope: the building's construction techniques and materials; 3. system: HVAC systems and renewable energy production; 4. operation: operational parameters impacting building utilization.

Furthermore, acquired data are categorized according to age, location, and type. Corgnati et al. (2013) [26] state that there are three methods for classifying RBs:

1.  "Example (Reference) Building". When no statistical data are available, this approach is applied, and it consequently depends on expert opinions and research. The end

result is a structure that is the most likely among a set of buildings in a given location and age.

2. "Real (Reference) Building". The RB is the most common building in a certain class. It is an actual, existent building with statistically average qualities.

3. "Theoretical (Reference) Building". This technique uses statistical data to define a reduced set of RBs as a statistical combination of the attributes observed within a building category in the stock. As a result, the structure is constructed using the most commonly used products and components.

The decision between these alternatives should be based on professional advice and the availability of statistical data. Different approaches can be used for various construction groups.

Building typologies are an effective tool for gaining a thorough understanding of the energy performance of various building forms and categories. Residential building typologies for seven European Mediterranean (MED) nations were evaluated in the scope of the HAPPEN project (H2020-EE-2016-2017) using a similar methodological approach. Each national typology comprises a categorization scheme that classifies buildings based on their size, age, and other energy-related criteria.

Determining the cost-optimal renovation scenario for each individual building is an almost impossible task to perform, and therefore, in order to overcome this difficulty, an RB can be used to reflect the typical and average building stock in a given nation. As a result, the developed RB should reflect the real national building stock as closely as possible so that the approach may produce accurate calculation results.

The essential data for each MED country addressed by the project were obtained from previous research and other EU initiatives, as well as from the expertise and experience of project partners. In circumstances where the necessary data cannot be retrieved, the use of default data is advised. This information is necessary for the corresponding clusters to be formed, for example, by building size (single/multi-family houses) and/or certain age bands, in order to be able to identify different subsets of building stock.

Based on past EU initiatives, as well as regional research and expert knowledge, each partner gathered and evaluated data addressing the general features of heating, DHW, cooling, construction, and geometrical details relevant to all types of residential structures.

The selection and formulation of relevant indicators is critical for properly monitoring energy saving efforts in building stocks. The indicator system must be appropriately chosen to map the status of the building stock at a certain time period as well as understand the dynamics of evolution. Basic and structural data are necessary for the establishment of building stock models for energy balance calculations.

According to all the above, it was decided to focus on three different construction periods (<1980, 1981–2000, 2001–2010) without taking into consideration buildings after 2010 and EPBD's issue because they may not need refurbishment. The division into three age groups can be viewed as a technique to simplify the overview, while it may hide considerable information. In particular, some of the recognized building types exhibit an overlap of the age categories, implying that one building type incorporates structures from other groups.

In this respect, the residential building types were reduced in order to minimize the noise created by complex definitions, errors, and misunderstandings. More specifically, single-family houses (SFH) will also include terraced houses, and multi-family houses (MFH) will include apartment blocks as well.

As a result, the RB can be defined as a structure that possesses representative qualities for the indicators listed below:

- Building construction type (as defined at the national level).
- Age (year of construction/commissioning).
- Geometry, including footprint type, total floor area, number of floors.
- Compactness, including wall/window area per orientation.
- Building energy systems and resources (e.g., HVAC, DHW, artificial illumination).

- Type of heating/cooling system, including fuel type and COP.
- Construction materials and thermal properties.

Therefore, the suggested methodology comprises the production of a virtual building using a hybrid approach, which is based first on fundamental statistical data and then on expert inquiries and other sources of information.

The added value of this study is the analysis of heterogeneous data sources, the collection and comparison of building stock information within a common comparative framework of building typology data between nations, and the contribution to the harmonization of the building typology approach.

The methodology implemented was as follows. Firstly, as we have seen in the introduction, we based the characterization properties in terms of energy consumption. Since the framework of this study is the HAPPEN project and we will look for cost-optimal consumption solutions by making specific improvements to the envelope, the building stock of the member countries was characterized by its geometry, the quality of its epidermis, and energy consumption. The definition of these necessary properties is based on the characterization of parameters on which the thermal performance of buildings depends. For the characterization of the building stock we focus on the geometrical characteristics and the characteristics of the thermal envelope. Later, these 42 reference buildings had their energy consumption quantified using simplified methods such as the one presented by Jara et al. [27].

As a result of these classifications we will find different levels of grouping. The first level of grouping is determined by the data provided by the member countries through the generated template. The second grouping method, as there may be some redundancy between buildings from different countries, is to establish a geometric and thermal classification in order to reduce the number of reference buildings to be considered.

The 42 reference buildings were then characterized in terms of energy consumption, and the energy consumption could be correlated with the classification in terms of geometry and envelope quality. Hierarchical clustering was implemented using Ward's minimum variation method [28] which was already used by Patteeuw et al. (2019) [29] previously. This clustering was performed using IBM SPSS software, establishing two levels of clustering.

In short, with this methodology we will obtain four types of groupings with different numbers of buildings depending on the degree of detail:

- Clustering Method 1: A total of 42 reference buildings obtained through a literature review. These buildings should be broken down in terms of geometry and envelope quality. They can then be implemented in the cost-optimal study through building improvements requested by the HAPPEN project.
- Clustering Method 2: A total of 37 reference buildings obtained through redundancy clustering of envelope geometries and qualities.
- Clustering Method 3: Nine reference buildings obtained through the hierarchical clustering study with Ward's minimum variation method. Ward's Dendogram of the first level.
- Clustering Method 4: Four reference buildings obtained through the hierarchical clustering study with Ward's minimum variation method. Ward's dendrogram in the second level

The procedure was as follows:

- Step 1: Study and definition of constructive properties.
- Step 2: Generation of forms to be filled in by member countries.
- Step 3: Data collection, analysis, and generation of reference buildings on Clustering Method 1.
- Step 4: Definition of scales based on U-values differentiating two groups: first, for roofs, slabs on grade, and walls, and second, for windows. Definition of scales: definition of scales based on construction geometry.

- Step 5: Classification of reference buildings on scales defined in step 4. Generation of reference buildings on the Clustering Method 2.
- Step 6: Calculation of annual energy consumption of the 42 reference buildings using the approximate method.
- Step 7: Implement Ward's minimum variation method and generate a dendrogram. Two levels of clustering, the second level with nine reference buildings and the first level (Clustering Method 3) with four reference buildings (Clustering Method 4).

First, the study and definition of constructive properties are necessary for the characterization of the building stock of the member countries.

Forms to be filled in by member countries were generated to request the necessary information. The forms will have three construction periods and two types of housing in each period, for a total of six buildings for each country. Through the collaboration of seven MED countries, the HAPPEN Project developed a harmonized structure for residential building typologies. A set of typical residential buildings was developed for each participant country, and data in terms of construction time and building type were collected. RBs are considered as examples and theoretical buildings according to the definition mentioned above. There are an endless number of permutations in the current building stock, and there is no "correct average". Therefore, the creation of a typical, in terms of size, number of floors, and use, RB for each building category (SFH–MFH) is recommended.

The majority of these input data for present and former building regulations and standards may be gathered from the sources indicated above for the definition of a building's geometrical characteristics. More specifically, the definition of their thermal characteristics can be achieved in several different ways. The related U-value criteria are stated since the building stock is classified into groups depending on the year of construction, which might correspond to the periods for changes in building regulations (or other changes of relevance for the assessments to be performed). RBs might be defined with U-values for each of these periods of time for each building category. The collected data structure for each MED country is depicted in Figure 1.

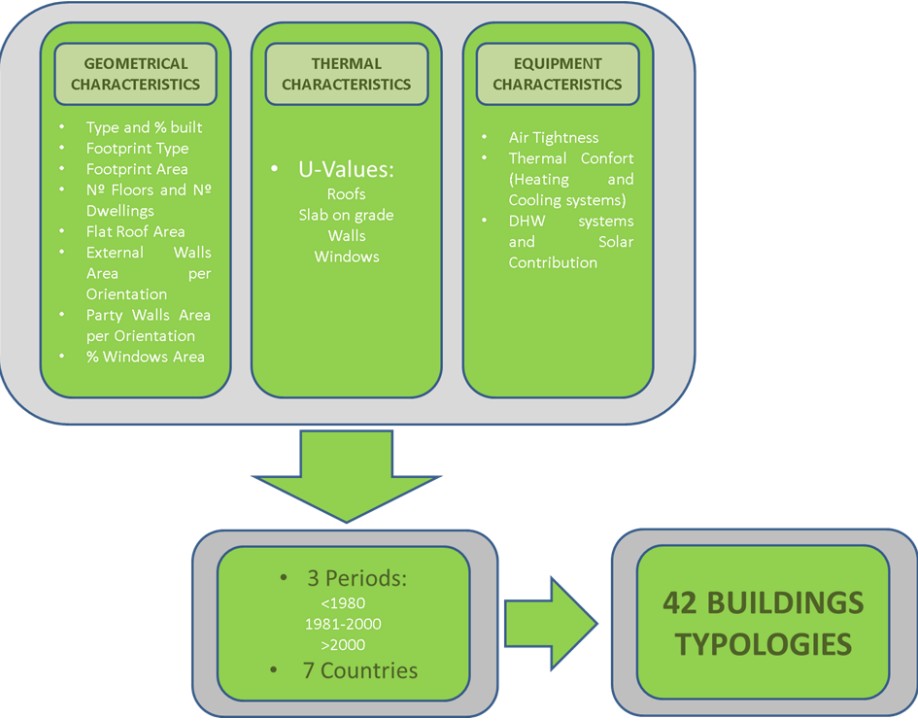

**Figure 1.** Data structure for RBs for each country per construction period.

Each existing real building demonstrates a different constructive nature; therefore, it is necessary to develop a classification with the objective of agglomerating the types that are congruent with each other. This method is called clustering. Descriptive indicators considered include:

- Geometrical characteristics, which define the dimensions of the envelope.
- Thermal characteristics, which define the construction materials and methods.
- Positional characteristics, which define the position of the building in relation to other buildings.

Data on thermal comfort and energy generation systems were requested, but for the clustering it was decided not to include the systems because the number of clusters increased considerably. For the application of cost-optimal renovation search, the software created for the HAPPEN project needs to be running for long periods of time. It is not feasible to include the systems from the beginning for this application. The systems will be included as soon as the building envelope is optimized.

Because we have six buildings per country (three building periods for two housing types, SFH and MFH) and seven member countries, a total of 42 reference building types are collected. This amount is not manageable for a mass pass study. Due to the framework of the project, these 42 buildings will be evaluated in terms of orientation (north, south, east, west), operation (number of cohabitants), and equipment (types of systems for thermal comfort), which can lead to an immense number of combinations for this purpose. In addition, there are buildings in different countries with similar geometries or similar thermal characteristics of the building envelope. These two reasons lead us to look for groupings that are coherent without losing relevant information. This first clustering generates the results of Clustering Method 1.

In order to carry out Clustering Method 2, two construction characteristics, thermal characteristics and geometrical characteristics, were taken into account.

Thermal characteristics will depend on the quality of roofs, slab on grade, walls, and windows. In order to be able to label the thermal characteristics, the code TXX is used. As provided in Table 1, each letter corresponds to a specific U-value range for roofs, slabs on grades, walls, and windows.

**Table 1.** Categorization of U-value ranges for walls, slabs on grade, roofs, and windows.

| Scale Based on U-Values (W/m$^2$ K) | | | |
|---|---|---|---|
| **Roofs, Slabs on Grade, Walls** | | **Windows** | |
| A | 0.01–0.49 | A | 0.90–1.39 |
| B | 0.50–0.99 | B | 1.40–1.89 |
| C | 1.00–1.49 | C | 1.90–2.39 |
| D | 1.50–1.99 | D | 2.40–2.89 |
| E | 2.00–2.49 | E | 2.90–3.39 |
| F | 2.50–2.99 | F | 3.40–3.89 |
| G | 3.00–3.49 | G | 3.90–4.39 |
| H | 3.50–3.99 | H | 4.40–4.89 |
| I | 4.00–4.49 | I | 4.90–5.39 |
| J | 4.50–4.99 | J | 5.40–5.89 |
| K | 5.00–5.49 | K | 5.90–6.39 |

Code TXX (Table 2) represents the number of the 11 thermal clusters which condensate the thermal properties of walls, slabs on grade, roofs, and windows, following the criteria given in the following tables. Letters (A, B, C, D, etc.) indicate the thermal properties of the various components of the building's envelope. As provided in Table 2, each letter corresponds to a specific U-value range for roofs, slabs on grades, walls, and windows. This grouping is made through the study of the thermal characteristics of each of the reference buildings provided by the member countries.

**Table 2.** Thermal clusters of walls, slabs on grade, roofs, and windows.

| | Roofs | Slabs on Grade | Walls | Windows |
|---|---|---|---|---|
| | | **Thermal Clusters** | | |
| T1 | F | C | C | J |
| T2 | B | B | B | E |
| T3 | C | E | D | H |
| T4 | A | A | A | D |
| T5 | A | A | A | B |
| T6 | B | C | C | D |
| T7 | A | C | A | C |
| T8 | E | E | C | I |
| T9 | G | D | C | K |
| T10 | G | G | G | H |
| T11 | G | G | B | G |
| T12 | C | B | B | E |
| T13 | C | B | D | D |
| T14 | B | D | D | G |

Geometric characteristics are broken down into floor area, building shape, number of floors, and position. To label the geometric features, the CYY code is used. In Table 3, code CYY illustrates the number of the scale based on construction geometry, which condensate the footprint area, the shape, the number of stories, and the number of party walls for the cases of SFH and MFH, respectively.

**Table 3.** Categorization of construction geometry for both SFH and MFH.

| | Scale Based on Construction Geometry |
|---|---|
| | **Single Family Houses (SFH)** |
| C1 | $\leq$150 m$^2$; $\leq$2 story; detached |
| C2 | $\leq$150 m$^2$; $\leq$2 story; semidetached |
| C3 | $\leq$150 m$^2$; 3 to 6 story; detached |
| C4 | $\leq$150 m$^2$; 3 to 6 story; semidetached |
| C5 | 150–600 m$^2$; $\leq$2 story; detached |
| | **Multi-Family Houses (MFH)** |
| C6 | 150–600 m$^2$; C-shaped; 3 to 6 story; terraced |
| C7 | 150–600 m$^2$; I-shaped; 3 to 6 story; detached |
| C8 | >1000 m$^2$; I-shaped; >6 story; detached |
| C9 | >1000 m$^2$; I-shaped; 3 to 6 story; detached |
| C10 | >1000 m$^2$; L-shaped; >6 story; terraced |
| C11 | 600–1000 m$^2$; I-shaped; 3 to 6 story; terraced |
| C12 | 150–600 m$^2$; I-shaped; $\leq$2 story; detached |
| C13 | $\leq$150 m$^2$; I-shaped; 3 to 6 story; terraced |
| C14 | 150–600 m$^2$; U-shaped; 3 to 6 story; detached |
| C15 | 600–1000 m$^2$; I-shaped; >6 story; detached |
| C16 | 150–600 m$^2$; I-shaped; >6 story; detached |

Once the partial classification has been carried out, in which the geometric and thermal categories of each of the buildings that make up the building stock of the member countries have been established, we prepare to make groupings in order to reduce the number of reference buildings. To achieve this, the buildings that were the same were reviewed and could be eliminated due to repetition. This second grouping generates the results of the Clustering Method 2 (Figure 2).

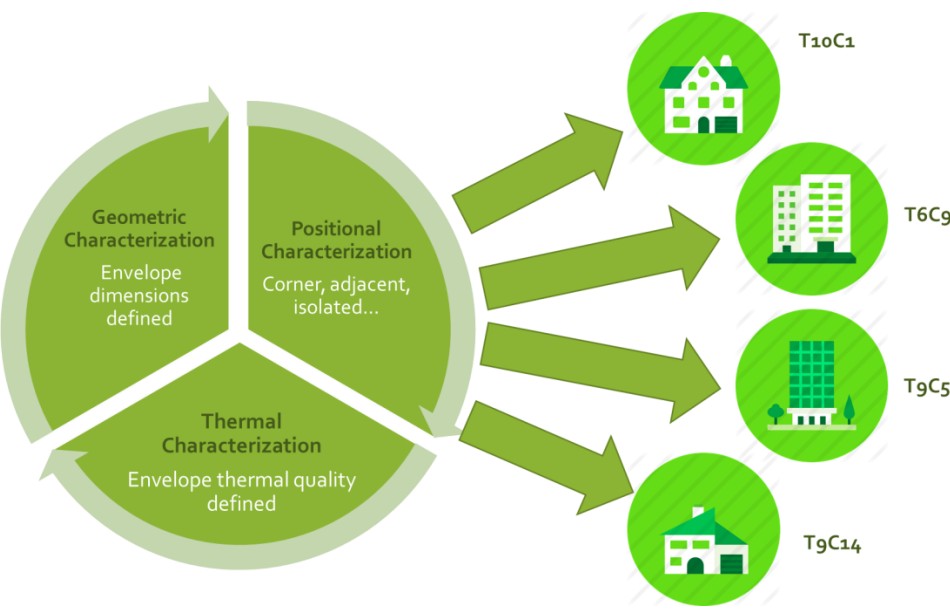

**Figure 2.** Clustering Method 2.

For the implementation of Clustering Method 3 and Clustering Method 4, the annual consumption per square meter of the 42 reference buildings was calculated using the approximate method developed by Jara et al. [27] (Table 4).

**Table 4.** Annual consumptions per square meter and dendrogram code.

| | | Annual Consumptions per Square Meters (kWh/m²·year) | | | | |
|---|---|---|---|---|---|---|
| **Country** | **Type** | **Consumption (kWh/m²·year)** | **Dendrogram Code** | **Type** | **Consumption (kWh/m²·year)** | **Dendrogram Code** |
| Spain | SFH 1 | 183.90 | 1 | MFH 1 | 72.03 | 22 |
| | SFH 2 | 97.43 | 2 | MFH 2 | 41.19 | 23 |
| | SFH 3 | 33.72 | 3 | MFH 3 | 17.63 | 24 |
| France | SFH 4 | 155.37 | 4 | MFH 4 | 102.82 | 25 |
| | SFH 5 | 47.03 | 5 | MFH 5 | 20.19 | 26 |
| | SFH 6 | 30.43 | 6 | MFH 6 | 15.92 | 27 |
| Slovenia | SFH 7 | 78.4 | 7 | MFH 7 | 83.73 | 28 |
| | SFH 8 | 60.32 | 8 | MFH 8 | 52.01 | 29 |
| | SFH 9 | 27.76 | 9 | MFH 9 | 8.18 | 30 |
| Italy | SFH 10 | 211.3 | 10 | MFH 10 | 83.68 | 31 |
| | SFH 11 | 203.52 | 11 | MFH 11 | 55.00 | 32 |
| | SFH 12 | 143.37 | 12 | MFH 12 | 30.62 | 33 |
| Croatia | SFH 13 | 190.39 | 13 | MFH 13 | 87.86 | 34 |
| | SFH 14 | 74.45 | 14 | MFH 14 | 39.85 | 35 |
| | SFH 15 | 39.87 | 15 | MFH 15 | 35.80 | 36 |
| Cyprus | SFH 16 | 385.63 | 16 | MFH 16 | 187.20 | 37 |
| | SFH 17 | 195.42 | 17 | MFH 17 | 122.72 | 38 |
| | SFH 18 | 67.34 | 18 | MFH 18 | 42.27 | 39 |
| Greece | SFH 19 | 421.32 | 19 | MFH 19 | 161.21 | 40 |
| | SFH 20 | 322.29 | 20 | MFH 20 | 91.12 | 41 |
| | SFH 21 | 103.47 | 21 | MFH 21 | 59.53 | 42 |

Once the consumptions have been quantified, we implement Ward's minimum variation method for the values obtained. The IBM SPSS software tool executes this algorithm by generating dendrograms. In this dendrogram, the reference buildings are grouped together, pointing to the most representative building in each group (Figure 3). A dendrogram of hierarchical clustering on ED data for the six RBs of the seven MED countries (42 different

buildings) is represented in the Y-axis. In the X-axis, Ward's number is shown. We can see the two missing groupings. Depending on the grouping level, we find nine reference buildings for the first level and four reference buildings for the second level.

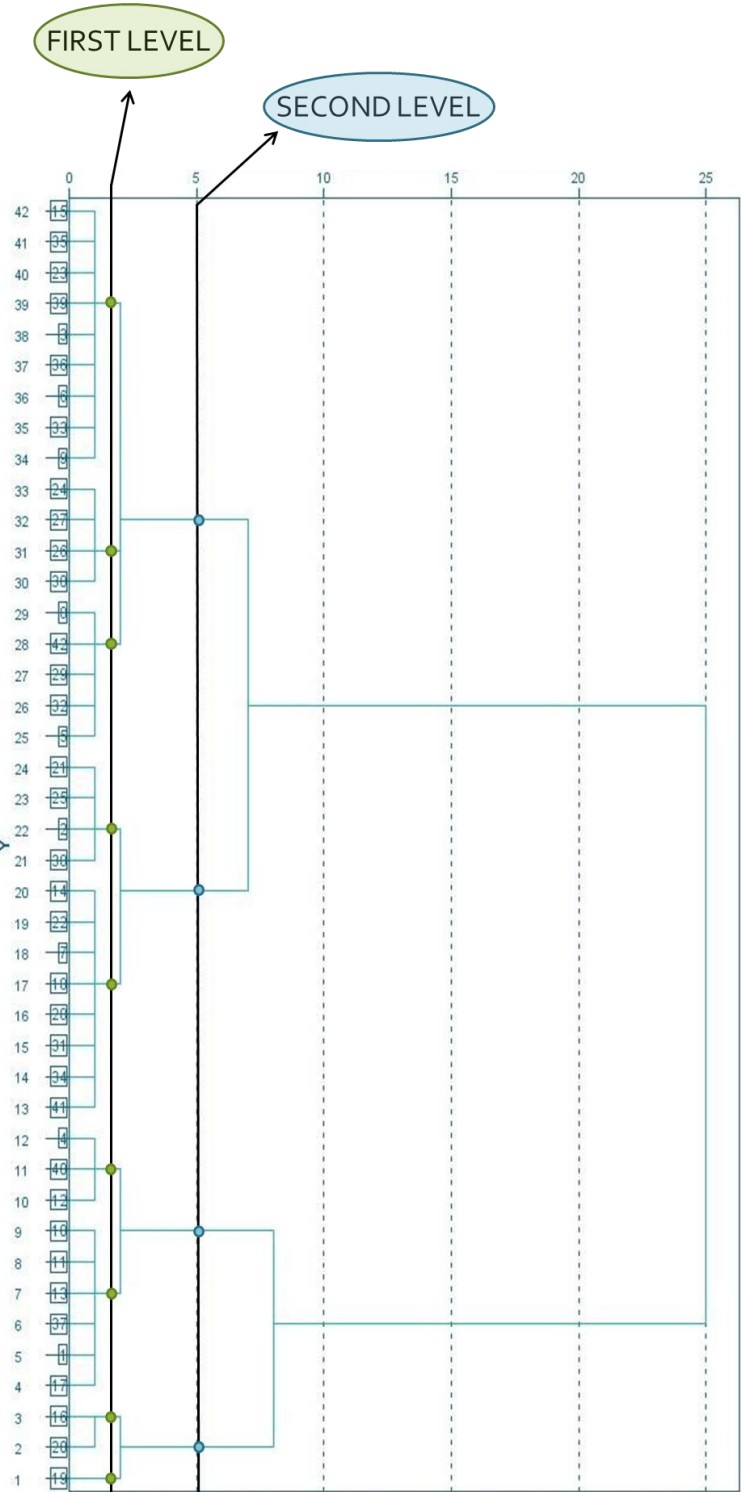

**Figure 3.** Dendrogram of hieratical clustering on ED data for the 6 RBs of the 7 MED countries (42 different buildings) represented in the Y-axis. In the X-axis, the Ward's number is shown.

## 3. Results and Discussion

Firstly, before starting the presentation of our results, it is crucial to show the findings in a usable manner. For this reason, we may divide the total into various probable applications:

- Useful reference buildings for evaluating the possibility of planned improvements to existing buildings such as evaporative cooling, intelligent control of air conditioning systems, and so on.
- Useful reference buildings for assessing the cost of intervention and the associated benefit following energy retrofitting of existing buildings.
- Useful reference buildings for estimating the cost of retrofitting existing buildings to meet stronger national rules aimed towards nZEB.

Once the potential uses are outlined, the various levels of cataloging reference buildings may have diverse applications. The grouping method 1 is appropriate for research with a high degree of detail, focusing on only one of the nations participating in this study, or focusing on only one building type. This is due to the fact that each country is represented by six reference buildings. Furthermore, by quantifying envelope properties, the use of these data in other works might give the option to conduct investigations with a high degree of detail. In the same way, Clustering Method 2 would be applied. The data acquired using this method can be used in studies that are required for reference buildings in the European Mediterranean region.

However, in case of implementing reference buildings in more general studies, such as those focused on the consumption derived from the quality and geometry of the envelope or in works with a large computational part, it would be more convenient to use the reference buildings derived from clustering methods 3 and 4.

It is important to note that the buildings derived from implementing Ward's minimum variance method are an approximation of agglomerating all the original reference buildings into a few buildings, allowing large-scale building improvement studies to be conducted without the number of possible cases making the task impossible in terms of calculation time.

In order to present our results in a clear way, we will focus on classifying them according to the previously mentioned clustering methods. First, a compilation of information provided by the member countries is displayed, with this grouping being the outcome of Clustering Method 1. Second, after implementing the previously described approach, the results of Clustering Method 2 are provided. Finally, the results from clustering methods 3 and 4 will be presented by using the Ward's minimal variance method.

- Clustering Method 1 results

After an exhaustive literature review, 42 reference buildings were generated from the seven member countries. The results are presented in Supplementary Materials, in Table S1. Thermal characterization of the building envelope of the Mediterranean–European building stock. The parameters studied are the type (MFH, SFH), the shape of the floor plan and whether it adjoins other dwellings, the floor area, the number of floors, the area of external walls, party walls, and windows broken down by orientation, and the U-values of each of the components of the thermal envelope.

- Clustering Method 2 results

Strongly linked variables generate important concerns in clustering analysis findings, and therefore variable reduction should be performed prior to clustering to eliminate redundant variables. In this research, five different variables (U-values, footprint area, shape, number of floors, and party walls) concerning the previous indicators were used. The clusters that were formed for each variable are presented in Table 5.

**Table 5.** Clusters per variable.

| Variables | U-Values | Footprint Area | Shape | Number of Floors | Party Walls |
|---|---|---|---|---|---|
| Number of Clusters | 11 | 4 | 4 | 3 | 3 |

The main objectives of this clustering are the following:

1.  Quantify the total number of different cases to be optimized.
2.  Identify among the previous cases the more representative ones and then verify their performance in the total number of cases.

In Table 6, one may notice 16 different typologies representing SFH and 21 MFH. This result was drawn due to the fact that three typologies of SFH are the same for different countries and time periods. This is the case for typologies T2C1 that are representative of Italy during the periods 1981–2000 and 2001–2010 and for Spain in 1981–2000. T6C5 is repeated for Slovenia during the periods <1980 and 1981–2000. Furthermore, T7C1 is found in the Croatian SFH during all three time periods. The typologies for a given country could be used to assess the optimal solutions in the reference climates for each country.

**Table 6.** Clustering Method 2—Total clusters for the examined MED countries.

| Period | Type | Country | | | | | | |
|---|---|---|---|---|---|---|---|---|
| | | Greece | Spain | Italy | Slovenia | Cyprus | Croatia | France |
| <1980 | SFH | T10C1 | T1C1 | T8C1 | T6C5 | T9C2 | T7C1 | T3C1 |
| | MFH | T10C7 | T10C6 | T8C7 | T6C9 | T9C14 | T3C9 | T10C16 |
| 1981–2000 | SFH | T11C1 | T2C1 | T2C1 | T6C5 | T9C5 | T7C1 | T4C1 |
| | MFH | T9C7 | T4C7 | T12C11 | T13C7 | T9C14 | T14C11 | T4C15 |
| 2001–2010 | SFH | T12C1 | T2C4 | T2C1 | T5C3 | T2C3 | T7C1 | T5C1 |
| | MFH | T2C7 | T5C10 | T7C12 | T5C8 | T2C9 | T4C7 | T5C11 |

- Clustering Method 3 results

In order to identify the more representative typologies, we employed hierarchical clustering with Ward's minimum variance method [28], which is a method that is not very common but has been used in recent studies, such as Patteeuw et al. (2019) [29]. "Hierarchical clustering is adopted since it leads to a single repeatable outcome" as a highly intriguing and viable strategy for grouping a building stock towards representative dwellings. This hierarchical clustering was performed using SPSS software with Euclidean distances (ED) and Ward's technique for linking. The number of clusters corresponds to the number of representative buildings in the aggregated model shown in Figure 2 as "cutting the cluster tree" at a specific value of Ward's linkage. Typically, those with a Ward's linkage value lower than five (first level) are considered representative clusters. In the present study, the buildings that can be considered really representative of the total population can be nine (for a Ward's number of two, as can be seen in Figure 3).

Table 7 summarizes the RB clusters obtained by using the aforementioned methodology. Each highlighted cell is the representative building of the cluster.

- Clustering Method 4 results

Implementing Ward's minimum variance method, "cutting the cluster tree" at Ward's linkage value of five (second level), the following reference buildings are generated. This scenario gives us four reference buildings. These RBs are summarized in Table 8.

Tables 9 and 10 identify which building is the most representative one of each cluster. As a result, we may obtain quantitative data on the building envelopes of each of the reference buildings obtained by the clustering methods used.

**Table 7.** Clustering Method 3—RBs in each cluster for a 9-cluster scenario.

| No. of RB | Representative Buildings in Each Cluster SFH/MFH Cluster Code from Table 6 | | | | | | | | |
|---|---|---|---|---|---|---|---|---|---|
| 1 | T2C9 | T7C1 | T14C11 | T4C7 | T2C4 | T4C7 | T5C1 | T7C12 | T5C3 |
| 2 | T4C1 | T5C10 | T5C11 | T5C8 | | | | | |
| 3 | T2C7 | T6C5 | T13C7 | T12C11 | T4C15 | | | | |
| 4 | T2C1 | T12C1 | T10C1 | T9C14 | | | | | |
| 5 | T2C3 | T7C1 | T10C6 | T6C5 | T6C9 | T8C7 | T3C9 | T9C7 | |
| 6 | T10C7 | T3C1 | T2C1 | | | | | | |
| 7 | T7C1 | T8C1 | T2C1 | T9C14 | T1C1 | T9C5 | | | |
| 8 | T9C2 | T11C1 | | | | | | | |
| 9 | T10C16 | | | | | | | | |

**Table 8.** Clustering Method 4—RBs in each cluster for a 4-cluster scenario.

| No. of RB | Representative Buildings in Each Cluster SFH/MFH Cluster Code from Table 6 | | | | | | | | |
|---|---|---|---|---|---|---|---|---|---|
| 1 | T2C9 | T7C1 | T14C11 | T4C7 | T2C4 | T4C7 | T5C1 | T7C12 | T5C3 |
| | T4C1 | T5C10 | T5C11 | T5C8 | | | | | |
| | T2C7 | T6C5 | T13C7 | T12C11 | T4C15 | | | | |
| 2 | T2C1 | T12C1 | T10C1 | T9C14 | | | | | |
| | T2C3 | T7C1 | T10C6 | T6C5 | T6C9 | T8C7 | T3C9 | T9C7 | |
| 3 | T10C7 | T3C1 | T2C1 | | | | | | |
| | T7C1 | T8C1 | T2C1 | T9C14 | T1C1 | T9C5 | | | |
| 4 | T9C2 | T11C1 | | | | | | | |
| | T10C16 | | | | | | | | |

**Table 9.** Representative building of Clustering Method 3 (reference Table S1).

| More Representative Building | Reference for Table S1 |
|---|---|
| T2C9 | Cyprus, MFH, 2000–2010 |
| T4C1 | France, SFH, 1981–2000 |
| T2C7 | Greece, MFH, 2001–2010 |
| T2C1 | Spain, SFH, 1981–2000 |
| T2C3 | Cyprus, SFH 2001–2010 |
| T10C7 | Greece, MFH, <1980 |
| T7C1 | Croatia, SFH, <1980 |
| T9C2 | Cyprus, SFH, <1980 |
| T10C1 | France, MFH, <1980 |

**Table 10.** Representative building of Clustering Method 4 (reference Table S1).

| More Representative Building | Reference for Table S1 |
|---|---|
| T5C11 | France, MFH, 2000–2010 |
| T7C1 | Croatia, SFH, 1981–2000 |
| T8C1 | Italy, MFH, <1980 |
| T11C1 | Greece, SFH, 1981–2000 |

## 4. Conclusions

The primary goal of this paper was to investigate the existing residential building stock in MED countries or regions represented by the HAPPEN project consortium, in order to assess key data concerning the general features and total energy consumption of residential buildings, allowing the definition of representative buildings as a basis for cost–benefit analysis.

The analysis began with the compilation of the essential information from past European and regional projects, publications, and reviews of related literature on the topic,

as well as other sources of statistical data or summarized information, in order to obtain a realistic picture of the residential building stock in the specific areas.

The gathering of information for each nation participating in the HAPPEN project was structured by using a thorough template that includes precise data essential for the identification of representative structures. Depending on the information available, a real or theoretical building was selected to provide performance data, which would identify the corresponding category.

Overall, the study objectives were met by identifying the most significant building classes (in terms of substantial renovation effect) to be further evaluated throughout the duration of the project. The paper provides information on selected building categories, providing extensive data for the majority of countries, which could be used as identification keys for the construction of "statistical representative buildings" or as checking indicators for the verification of an actual building chosen as an RB for a specific category (e.g., average conditioned area, compactness ratio, number of floors, EP indicators, age, systems, fuel type, etc.).

For the selected RBs in each MED country, the corresponding necessary geometrical data, building energy use, and base heat supply regime (type of the heating system, energy resource/carrier, etc.) should be included in the simulation of the energy consumption and the estimation of different major renovation strategies and packages of solutions.

The different levels of clustering lead to a variety of applications. For the application of mass runs (as we have mentioned, these reference buildings will be studied in the HAPPEN project), a high clustering level will be needed. Analyzing nine or four reference buildings derived from clustering methods 3 and 4, it would be sufficient to carry out renovations of the buildings with the method used in the framework of the project that encompasses this work.

For building renovation analysis work in only one participating country, we can use the data provided by the Clustering Method 1. Six reference buildings will be analyzed, allowing cost-optimal studies to be carried out without being disproportionate.

In applications of feasibility studies of thermal comfort systems, they can be carried out on the 37 buildings of Clustering Method 2 or on the 42 buildings of Clustering Method 1, making a complete study on the optimization of equipment dedicated to thermal comfort.

The same applies to the analysis of electrical and thermal power generation equipment. For a feasibility study of generation equipment, or cost-optimal studies, the 42 reference buildings or the 37 buildings of clustering methods 1 and 2 can be used.

**Supplementary Materials:** The following supporting information can be downloaded at: https://www.mdpi.com/article/10.3390/app12115638/s1. Table S1. Thermal characterization of the building envelope of the Mediterranean–European building stock. References [30–55] are cited in Supplementary Materials.

**Author Contributions:** Conceptualization, C.E. and J.M.S.L.; methodology, N.B. and M.R.; software, M.R. and P.T.; validation, N.B., P.T. and C.E.; formal analysis, C.E. and M.R.; investigation, C.E.; resources, C.E.; data curation, P.T.; writing—original draft preparation, C.E., N.B. and J.M.S.L.; writing—review and editing, C.E., N.B. and J.M.S.L.; visualization, C.E.; supervision, M.N.A.; project administration, M.N.A. and J.M.S.L.; funding acquisition, M.N.A. and J.M.S.L. All authors have read and agreed to the published version of the manuscript.

**Funding:** This research has received funding from the European Union's Horizon 2020 research and innovation program under Grant Agreement No 785072—"Holistic APproach and Platform for the deep renovation of the med residential built Environment" HAPPEN Project.

**Institutional Review Board Statement:** Not applicable.

**Informed Consent Statement:** Not applicable.

**Data Availability Statement:** The data presented in this study are available upon request from the corresponding author. The data are not publicly available due to privacy restrictions.

**Acknowledgments:** The authors would like to thank the European Union's Horizon 2020 research and innovation programme. Also, the authors wish to give thanks to all country members for collaborate in recompilation of data.

**Conflicts of Interest:** The authors declare no conflict of interest.

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
