# Peer review of "Identification of Reference Buildings in Mediterranean Countries: The HAPPEN Project Approach"

_applsci, doi:10.3390/app12115638_

Round 1

Reviewer 1 Report

In the reviewer's opinion, the structure of the article is very disturbed.

The novelty of the authors' efforts should be apparent from the introduction.
The methodology presents a literature review on research methods in the thematic scope of the article and some information that should probably be included in the introductory part.
The results section presents partial information on the chosen methodology and excerpts from the research results. There is no indication of a clear method of calculation, numerical data of their statistical processing.
In the conclusions, a summary of the article is presented.

The subject matter is interesting, but the research presentation has serious flaws and in the reviewer's opinion does not allow the article to be published in Applied Sciences journal.

Reviewer 2 Report

This study investigates the residential building stock in seven Mediterranean countries or regions represented by the HAPPEN project consortium in order to define representative buildings for subsequent energy analysis. However, the aim of this research is not clearly specified in the introduction section.

The paper deals with an important topic. Nonetheless, references in the manuscript are not too recent. I encourage authors to update the manuscript with the latest research, such as:

  • Li, X., Yao, R., Liu, M., Costanzo, V., Yu, W., Wang, W., … Li, B. (2018). Developing urban residential reference buildings using clustering analysis of satellite images. Energy and Buildings, 169, 417–429. https://doi.org/10.1016/j.enbuild.2018.03.064
  • Bhatnagar, M., Mathur, J., & Garg, V. (2019). Development of reference building models for India. Journal of Building Engineering, 21(October 2018), 267–277. https://doi.org/10.1016/j.jobe.2018.10.027
  • Ledesma, G., Pons-Valladares, O., & Nikolic, J. (2021). Real-reference buildings for urban energy modelling: A multistage validation and diversification approach. Building and Environment, 203(March), 108058. https://doi.org/10.1016/j.buildenv.2021.108058
  • Foroushani, S., Bernhardt, R., & Bernhardt, M. (2022). On the use of the reference building approach in modern building energy codes. Energy and Buildings, 256, 111726. https://doi.org/10.1016/j.enbuild.2021.111726

In line 183 authors decided “to focus on three different construction periods (<1980, 1981-2000, 2001-2010) without taking into consideration buildings after 2010 and EPBD’s issue because they may not need refurbishment”. Nonetheless, regulations on thermal requirements came into force in different years, in the seven analysed countries. Given that the clustering takes thermal aspects of the buildings into account, authors could consider focus on thermal requirements rather than on construction year.

To improve understanding of the research, the methodology section should include the clustering methodology described in section “3 Results and Discussion”.

Table 5 describes the T10C1 and T4C1 cluster for MFH (France). However, in Table 4, C1 corresponds to SFH. What is the explanation?

Results obtained should be discussed in more depth in the section “3 Results and Discussion”. For example: authors could detail the meaning and implications of the selected clusters…

The quality of Figure 2 is very poor, and hardly readable. Please increase the image quality.

The representative buildings selected in Table 7, it seems that they do not correspond to those identified in Figure 2. However, the quality of Figure 2 does not allow its verification. Please could you verify?

Reviewer 3 Report

The paper presents an interesting topic but a clear structure is missing, it needs a major revision in other to be publish. Methods don’t describe the procedure followed, whereas the section results contents many information that should be described in the methods section. Section 3, results and discussion, doesn’t content any discussion about the results. Results are only partially shown but not discussed. Additionally, the clustering performed only include some of the collected parameters from Figure 1 and it is not clear why important parameters such as the type of heating system is not included, this should be part of the results and discussion section. Conclusions section has to be completely reformulated, what it is presented at the moment is a summary of the work performed and not conclusions from the results, such as: allowed the methods employed to reach the goal of the research? Which results can be highlighted? And so on. Additional commentaries:

Line 79: delete stocks after residential buildings.

Introduction: many different methodologies to determine RB are presented, however, a critical analysis is missing: what are the similarities and differences between them. It is neither Clair what the authors have taken from this analysis, the last paragraph of the section is in this sense weak. Additionally, in the introduction it should be clearly stated why or in which way the presented research goes beyond the state of the art or in which way improves the existing knowledge. Lines 129-130: can this be considered three factors? What the authors call building type is rather building use and it has not relation to the thermo-physical characteristics of the building so it has to be considered a different factor. Isn’t building use a better term than building type? Line 162: do you mean cost-optimal renovation scenario? Lines 162-166: here you are talking about RBs, multiple reference buildings, or? In other case is imposible to reflect a national building stock only with one building. Please correct accordingly. Methods: methods are not clearly described, information given us too loose and a clear structure is missing. The followed steps have to be clearly described and the decisions taken explained. In the lines 207-210 the authors presents what is the novelty of the paper however these aspects are not covered by the methods. These have to be described in the steps followed to conduct the presented research. Line 204: I think the hybrid approach is quite common, in many cases statistical information is completed with experts knowledge. Lines 212-227: this content corresponds to the methods. Lines 231-243: clustering is a method therefore it should be explained in the method section. It is missing which role plays clustering within the definition of the RBs. Table 4: it is not clear how party walls are included in the clusters, these do not appear in the tables. Lines 256-257: please be more precise in the description. The combination of the thermal clusters with the construction clusters leads to the total clusters. Total cluster is also a bad nomenclature, typology cluster won’t be clearer? Lines 269-270: how is it possible that one typology covers the three periods? I can understand that the construction typology can be the same but the U-Values can be only the same if most of the buildings have already been renovated to reach that thermal standard. Lines 273-282: these lines describes the method used. Conclusions is section 4 from the paper.

Reviewer 4 Report

Identification of reference buildings in Mediterranean countries: The HAPPEN project approach 

This article is well structured with clear aims and objectives.

As all countries are part of the EU, the scope of this article should clearly state this.

In the research methodology section, the authors have clearly illustrated the sources of information and the data selection which is brilliant, I would recommend that they expand on how these data were analysed.

The analysis is very thorough and the findings are presented using tables and a diagram. The latter was not clear, however.

Also, it would be better if the real and theoretical building patters/types were clearly highlighted and justified.

As a general note, while I totally agree on the novelty of this manuscript, in terms of aims and approach, I am not sure about the generalisation of findings to the 7 selected nations. Maybe a paragraph or a section to acknowledge the locality and different aspects would have help to keep this in mind.

Also, this might be useful to be acknowledged as part of the limitation of this research in the conclusion.

Reviewer 5 Report

The paper concerns the configuration of reference buildings in Mediterranean countries. This subject was popular many years ago, especially around 2010 when the member states of EU were preparing their reports for the cost optimal levels of energy performance of buildings. For this study, every country defined reference buildings with respect to the use, the age, the architectural morphology, the geometry, the climate, the thermal characteristics, the systems, etc. The present paper has nothing to add to these studies. There is no novelty in the approach and the results are expected.

The source of the data used in the categorization of the reference buildings are very vaguely mentioned. To my mind, previous studies/papers/etc. are not adequate for such a study; the actual characteristics of the building stock categories has to found through a very thorough research, conducted by specialized scientists, working with buildings, their design and their components. The fact that the authors didn't conduct such a study lead to errors in their approach; for example, the U values of the last construction period should be significantly lower than the ones reported. [ and a minor correction: "glazing" should be replaced by windows or fenestration, as the transparent building element is considered as a whole and the glazing is only one of its components.] 

Another important shortcoming is that the results of the study are not described in detail and thus they cannot be exploitable by other researchers. Still, making categories and clusters do not correspond to a building structure that could be used in order to define appropriate measures for its energy performance improvement in a reliable way, given that the calculation tools, even the simpler ones, require specific data that are not addressed in the study. 

Good points of the paper are the use of language and the presentation of the state of the art (although there are still relevant studies to mention),

Round 2

Reviewer 1 Report

The article has improved considerably since the last version.
Given the stylistic errors that occur, I recommend rereading it and making linguistic corrections.
The introduction of illustrative elements, for example, diagrams instead of empty tables indicating headline data would improve the communication of important content.
Below are the specific corrections that should be made.
-L. 275-276: Sentence should be rebuilt or explained.
-Figure 1. should be improved. I would suggest to show the headings data in the form of the diagram without empty cells of the table.
-L. 315: Translator advertisement should be deleted.
-L. 335: and further: shortcuts ike CYY should be explained.
-L. 409: repeated text "table 6" should be deleted.

Yours Sincerely,

Reviewer.
